# Albumin Stabilized Fe@C Core–Shell Nanoparticles as Candidates for Magnetic Hyperthermia Therapy

**DOI:** 10.3390/nano12162869

**Published:** 2022-08-20

**Authors:** Maria Antonieta Ramírez-Morales, Anastasia E. Goldt, Polina M. Kalachikova, Javier A. Ramirez B., Masashi Suzuki, Alexey N. Zhigach, Asma Ben Salah, Liliya I. Shurygina, Sergey D. Shandakov, Timofei Zatsepin, Dmitry V. Krasnikov, Toru Maekawa, Evgeny N. Nikolaev, Albert G. Nasibulin

**Affiliations:** 1Skolkovo Institute of Science and Technology, 3 Nobel Street, 121205 Moscow, Russia; 2Hi-QNano s.r.l., Via Barsanti No. 1, 73010 Arnesano, Italy; 3Department of Engineering of Innovation, University of Salento, Via per Arnesano km 1, 73100 Lecce, Italy; 4School of Chemical Engineering, Aalto University, Kemistintie 1, 02015 Espoo, Finland; 5Graduate School of Interdisciplinary New Science, Toyo University, Kawagoe 350-8585, Saitama, Japan; 6Bio-Nano Electronics Research Center, Toyo University, Kawagoe 350-8585, Saitama, Japan; 7V.L. Talrose Institute for Energy Problems of Chemical Physics at Federal Research Center of Chemical Physics, Russian Academy of Sciences, Leninsky Prospect 38 Building 2, 119334 Moscow, Russia; 8Kemerovo State University, Krasnaya 6, 650043 Kemerovo, Russia

**Keywords:** ferromagnetic particles, core–shell nanoparticles, magnetic hyperthermia, iron nanoparticles, flow-levitation method

## Abstract

Carbon-encapsulated iron nanoparticles (Fe@C) with a mean diameter of 15 nm have been synthesized using evaporation–condensation flow–levitation method by the direct iron-carbon gas-phase reaction at high temperatures. Further, Fe@C were stabilized with bovine serum albumin (BSA) coating, and their electromagnetic properties were evaluated to test their performance in magnetic hyperthermia therapy (MHT) through a specific absorption rate (SAR). Heat generation was observed at different Fe@C concentrations (1, 2.5, and 5 mg/mL) when applied 331 kHz and 60 kA/m of an alternating magnetic field, resulting in SAR values of 437.64, 129.36, and 50.4 W/g for each concentration, respectively. Having such high SAR values at low concentrations, obtained material is ideal for use in MHT.

## 1. Introduction

Magnetic nanoparticles (NPs) are now actively applied in various technological fields, including data storage, catalysis [1], MRI contrast agents [2], and drug delivery [3]. Nevertheless, magnetic hyperthermia therapy (MHT), a complementary cancer treatment method, is one of the most promising techniques [4,5]. MHT employs the heat generated by a magnetic nanoparticle subjected to an alternating magnetic field; the heat, consequently, induces local cytotoxic effects such as denaturation of cytoplasmic and membrane tumor proteins, reduction in blood flow, and induction of tumor acidosis. Hyperthermia also amplifies the immune responses in the body against cancer while decreasing the immune suppression and immune escape of cancer [6,7]. Specific absorption rate (SAR) is a key performance parameter marking the applicability of a material [8] and determining the dose and duration of the MHT [9]. The high SAR depends on numerous parameters of NPs: size, shape, composition, magnetic interaction, concentration, as well as the applied magnetic field frequency and strength [10].

Various magnetic compositions, including spinel-based MFe_2_O_4_ (M = Zn, Mn, Co, Ni), NiCo_2_O_4_, and perovskite La_1−x_Sr_x_MnO_3_, La_1−x_Sr_x_Mn_1−y_Fe_y_O_3_ structures have been already applied for hyperthermia cancer treatment [11,12,13,14,15,16]. Nevertheless, iron oxide nanoparticles—magnetite (Fe_3_O_4_) and maghemite (γ-Fe_2_O_3_)—are the most widely studied and show superior biocompatibility in comparison to Co and Ni-containing compounds [17,18,19]. However, the γ-Fe_2_O_3_ phase is metastable and, therefore, can be easily transformed into antiferromagnetic α-Fe_2_O_3_; moreover, the presence of different types of oxide in the sample may deteriorate SAR [20].

Carbon-encapsulated magnetic metal nanoparticles are another type of nanostructured carbon material. These nanoparticles have unique magnetic and electrical properties, as well as high thermal and chemical stabilities due to nanoparticle protection by a graphitic layer. Essentially metallic materials based on Ni, Fe, Co, and their alloys demonstrate an excellent magnetization. Among them, iron is the most promising one since it displays a colossal magnetization [21], biocompatibility, and shows the highest absorption rates in comparison to the complementary oxide forms [22].

Carbon encapsulated iron NPs (Fe@C) can be synthesized by a wide range of approaches: pulsed plasma method [23], iron carbonyl Fe(CO)_5_ pyrolysis [24], high-energy ball milling of dopamine and ferric nitrate [25], hydrothermal carbonization of iron chloride FeCl_2_ [26], carbon arc discharge [27], etc. However, they are time-consuming, energy-demanding, or require sophisticated laboratory equipment. Synthesis of carbon encapsulated structures is usually accomplished under extreme conditions by using high voltages and temperatures and mostly demonstrated low yields of the desired product.

The Guen–Miller flow–levitation method is an alternative approach that allows the synthesizing of core–shell nanoparticles [28], including Fe@C in a high yield [29]. This technique is based on the evaporation of metal from the surface of a melted metal droplet levitating in the high-frequency electromagnetic field of a countercurrent inductor in a vertical reactor. Metal nanoparticles are formed due to the transfer of metal vapors in an inert gas flow to the condensation zone, where they interact with a carbon source in the gas phase downstream to form core–shell structures. This method is the most promising for synthesizing fine powders and magnetic nanoparticles, as it features a high production rate and outstanding uniformity of Fe@C core–shells [29].

To utilize core–shell particles in biomedical applications, including magnetic hyperthermia therapy, we need to prepare their stable dispersions in a non-toxic media [30]. Such dispersions are obtained by covalent or non-covalent functionalization of the carbon shell surface [31,32,33,34]. Common coatings are based on some of the most abundant proteins—human serum albumin (HSA) and bovine serum (BSA) albumin [35,36,37,38,39]. HSA and BSA are water-soluble, biodegradable, non-toxic, and efficient in stabilizing nanoparticles [35,39]. Therefore, albumin-coated Fe@C nanoparticles have immense potential in biomedicine.

In the present work, we synthesized core–shell iron nanoparticles (Fe@C) with a mean diameter of 15 nm by the flow–levitation technique using acetylene as a carbon source, and subsequently functionalized the particles with bovine serum albumin. The advantageous carbon shell protects the iron core from oxidation enhancing chemical stability and maintaining electromagnetic properties of the nanosystem. The electromagnetic properties of those BSA-coated Fe@C nanoparticles are reported for the first time. Such BSA-coated Fe@C nanoparticles demonstrate much higher saturation magnetization and SAR when compared to other types of commonly used magnetic nanomaterials based on iron oxide NPs. Further, we correlate the efficiency of the produced NPs for the MHT and discuss the effect of metal nanoparticle concentration on the efficiency.

## 2. Materials and Methods

Carbon-coated iron nanoparticles (Fe@C) were synthesized by evaporation–condensation flow–levitation method from iron wire using argon as a carrier gas and acetylene as a carbon source. A more detailed procedure is described elsewhere [40]. Briefly, iron seeds are nucleated from supersaturated iron vapor after preheating iron source (iron wire) in a high-frequency electromagnetic field (440 kHz). Further, formed iron nanoparticles catalyze decomposition of acetylene injected into argon flow below in the flow. As-synthesized Fe@C nanoparticles were collected on a filter and subjected to passivation by atmospheric air using two cycles “slow inflow followed by pumping out” [40].

For their stabilization in water, 1.0, 2.5, and 5.0 mg of Fe@C core–shell nanoparticles were dispersed in 1 mL aqueous solution of 1% Bovine Serum Albumin (BSA, Sigma Aldrich, Saint Louis, MO, USA) using an ultrasonic homogenizer (UH-600S, SMT Co., Ltd., Tokyo, Japan) for 40 min in a continuous mode. The samples were washed several times with deionized water and collected by centrifugation. For dynamic light scattering (DLS) and Zeta potential measurements, samples were re-dispersed in 10 mL sodium phosphate buffer with pH = 7.4. Zeta potential and particle size distribution of the BSA stabilized magnetic dispersion were estimated using DLS analysis Malvern Zetasizer Nano-ZS (Malvern Panalytical, Malvern, United Kingdom) equipped with HeNe laser, wavelength λ = 632.8 nm. All the measurements were conducted at 25 °C and diluted 10 times before analysis.

Nanoparticles were studied by means of FEI Tecnai G2 F20 transmission electron microscopy (TEM; FEI Company, Hillsboro, OR, USA) at an acceleration voltage of 200 kV and X-ray diffraction analysis (XRD) using Bruker D8 ADVANCE diffractometer (Bruker ASX GmbH, Karlsruhe, Germany) Cu Ka irradiation.

Magnetic measurements were carried out using a Quantum Design SQUID MPMS3 magnetometer (Quantum Design, Inc., San Diego, CA, USA). The zero field-cooled and field-cooled (ZFC/FC) curves were obtained with an applied magnetic field of 100 Oe in the temperature range of 2–300 K.

Heating experiments were conducted by locating the sample in the center of a copper coil, powered by an induction heating system (EasyHeat, Ambrell Ltd., Cheltenham, United Kingdom) and using a DC inverter chiller (RKE2200B1-V, Orion Machinery Co. Ltd., Suzaka, Japan). Measurements were performed and recorded at the magnetic field of 60 kA/m and the applied magnetic field frequency of 331 kHz. The temperature increase was monitored through a thermal camera and recorded with an acquisition time of 1 s. SAR values were estimated as [41,42,43,44,45,46]:(1)$$SAR=\left\{\left(m_{MNP}+m_{water}\right)/m_{MNP}\right\}\times C\times \left(\Delta T/\Delta t\right)$$
where $m_{MNP}$ and $m_{water}$ are the mass in grams of the magnetic nanoparticles and water, respectively. *C* is the specific heat capacity of the solution in J /g °C; Δ*T* is the temperature change; Δ*t* is the time interval in sec. For SAR calculation, the initial slope method was used, where $\frac{\Delta T}{\Delta t}$ is the slope of the first 60 s of magnetic field exposure [47]. Considering the specific heat values *C_Fe_* = 0.44 J /g °C [42] and *C_water_* = 4.18 J /g °C, it is possible to assume that $C=C_{water}$ by satisfying the following relation $C_{MNP}\ll \;C_{water}$.

## 3. Results and Discussion

### 3.1. Fe@C Structure

The morphology of the as-obtained Fe@C NPs was investigated by TEM (Figure 1). The powder (Figure 1a) mainly consists of spherical shape nanoparticles in a wide range from 6 to 26 nm, with a mean diameter of Fe@C about 15 ± 4 nm (Appendix A) with shell thickens c.a. 2.5 nm (Figure 1b).

The phase composition of the synthesized powder was determined using X-ray diffraction. XRD pattern (Figure 2) shows two phase systems, containing α-Fe (BCC) [48,49] and γ-Fe (FCC) phases. The formation of a high-temperature phase (γ-Fe) is associated with high cooling rates of particles, achieving 104–105 deg/s. The absence of a pronounced graphite diffraction peak at 2θ = 26.3°, corresponding to the (002) plane symmetry of graphene lattice (Figure 2) indicates that the produced carbons are mainly amorphous. The shell does not exhibit a layered graphitic structure (Figure 1b) and might be a mixture of carbide phase, amorphous carbon, and iron oxide. Meanwhile, in the process of passivation of the synthesized nanoparticles by air, the oxidation of the shell occurs with a formation of a small fraction of ferrimagnetic iron oxide [41].

### 3.2. MHT Performance

The magnetization of Fe@C magnetic nanoparticles is shown in Figure 3a. The curve appears nonlinear and has reversible characteristics demonstrating ferromagnetic behavior. The saturation magnetization value of the samples produced is Ms = 83.6 emu/g with a coercive force of Hc = 623 Oe. The Fe@C magnetization value is smaller in comparison to the bulk α-Fe phase [45] and might be associated with several factors which undermine the role of the external magnetic field presence of diamagnetic contribution of carbon shell, ferromagnetic Fe_3_C, and paramagnetic γ-Fe phases, and amorphous phases with a lower saturation magnetization or paramagnetic phase based on FeC_x_ [40]. The percentage of superparamagnetic fraction in the sample is around 30% according to the calculated values of coercivity for the core–shell iron nanoparticles with a mean diameter of 20 nm [50].

Temperature dependences of the zero-field-cooling and field-cooling magnetizations (M_ZFC_ and M_FC_, respectively), in the Fe@C sample in the whole temperature range of 2–300 K with splitting between M_FC_ and M_ZFC_ below 300 K are given in (Figure 3b). This behavior can be explained by thermally activated ferromagnetic moments of Fe@C. Pre-cooling the sample to 2 K leads to a chaotic distribution of magnetic moments. Further increase in temperature leads to an increase in the net magnetization due to the alignment of the ferromagnetic moments in the direction of the magnetic field. By the end of the cooling process, the ferromagnetic moments remain blocked in the direction of the applied field. The M_ZFC_ curve indicates an incomplete maximum within the measurement’s temperature limit, meaning that blocking temperature (TB) lies above 300 K at the applied field of 100 Oe. On the other hand, magnetization decreases faster below 80 K. It might be due to the fact that the blocking temperatures of the smaller particles are near 80 K, while their magnetic moments are blocked below 80 K, and also with being the formation of the spin glass state for Fe_3_C at 75K [51,52,53].

The surface charge of a particle typically determines its colloidal stability as well as its interactions with biological entities. Nanoparticles can form clusters in blood flow and stick to or interact with the oppositely charged cell membrane [54,55]. Stabilization of Fe@C nanoparticles by BSA solution results in negatively charged albumin-stabilized nanoparticles, making it possible to minimize nonspecific contact through electrostatic interactions with innately negatively charged blood components such as plasma and blood cells. The relationship between the concentration of the magnetic phase and the final albumin-stabilized particle size is shown in Table 1. An increase in the concentration of the magnetic phase from 1 to 5 mg/mL leads to an increase in the hydrodynamic diameter in the Fe@C-BSA system from 17 to 24 nm.

The zeta potential represents the surface charge of nanoparticles and reflects their long-term stability. The ZP value exceeding 30 mV, irrespective of their sign is considered to be the optimum for stabilization [56]. The −32 mV surface charge of obtained Fe@C-BSA dispersions assume their good stability. Indeed, neither formation of aggregates nor increase in particle size are observed within 6 months of the dispersion storage.

Clinical practice defines several limitations for MHT since MNPs can exhibit cytotoxicity at high concentrations. A previous study [57] indicated that the concentration of magnetic fraction should not exceed 5–10 mg/mL. Therefore, to minimize the cytotoxic effect, we fixed the maximal concentration of 5 mg/mL magnetic phase.

For SAR calculations, heating regimes for every concentration were measured at the magnetic field of H = 60 kA/m and frequency of ƒ = 331 kHz during 60,000 ms. The temperature variation as a function of measuring time (t) for BSA-Fe@C at different concentrations demonstrates a linear trend (Appendix A).

Our experiments were based on the following assumptions: 1. the sample temperature is homogeneous while heating upon an alternating magnetic field is applied; 2. heat losses are negligible during a certain period of time. The SAR values it is considered to be directly dependent on the concentration used, there are studies that show differences and even opposite behavior [58]. All the obtained values from the slope are expected for ideal adiabatic conditions with minimal heat losses in the environment. Even at low frequencies (kHz), the overall heat loss is relatively small and can be neglected [59]. The highest SAR value for BSA-Fe@C was obtained at the lowest concentration (Table 2).

For minimal invasiveness for a patient, it is necessary to obtain the highest SAR values at the lowest NP concentration. In the present work, a Fe@C concentration of 1 mg/mL allows the SAR value of 437.64 W/g. Although other systems might exhibit close SAR values, they are reached at higher concentrations that could have a high probability of the cell-mediated response (Table 3).

On the contrary, an alternating magnetic field plays an essential role because at the frequency and amplitude product *(H × f*) higher than 10^9^ A m^−1^ s^−1^ one can damage the surrounding tissue [60,61]. The nanoparticles with the highest SAR were measured at *H × f* values exceeding the optimal therapy conditions. In our case, samples were measured at 6 × 10^8^ A m^−1^ s^−1^, suggesting a better fit for existing procedures.

## 4. Conclusions

Fe@C core–shell nanoparticles were synthesized by the flow–levitation method. The obtained Fe@C nanoparticles feature a mean diameter of 15 ± 4 nm with a shell thickness of 2.5 ± 0.6 nm. The produced nanoparticles demonstrate ferromagnetic behavior with the saturation magnetization value of M_s_ = 83.6 emu/g. Stabilization of the Fe@C nanoparticles in BSA solution results in a stable dispersion without sedimentation with a ZP value around −32 mV for 1 mg/mL of Fe@C nanoparticle concentration. We evaluated SAR of Fe@C-BSA nanoparticles at the magnetic field of H = 60 kA/m and frequency of ƒ = 331 kHz to achieve the highest value of 437.64 W/g for a concentration of Fe@C-BSA of 1 mg/mL. The obtained value is optimal for MHT treatment since it does not exceed any of the established clinical limits and also a high SAR value at lower concentrations is ideal for therapy; thus, BSA-coated Fe@C nanoparticles are a promising material for implementation in MHT.

## Figures and Tables

**Figure 1 nanomaterials-12-02869-f001:**
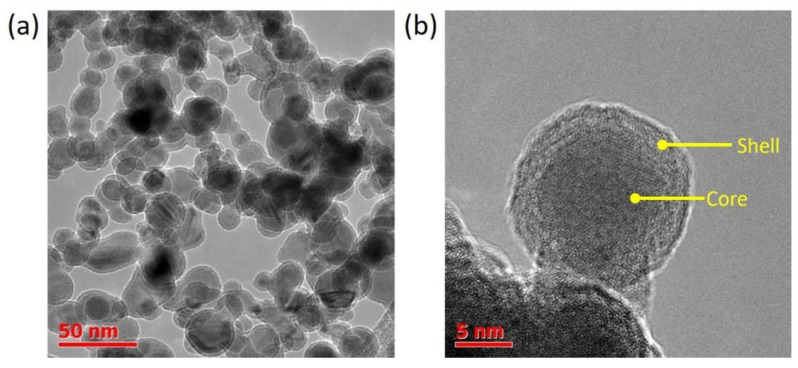
Representative TEM images of Fe@C particles: (**a**) overview and (**b**) close-up view.

**Figure 2 nanomaterials-12-02869-f002:**
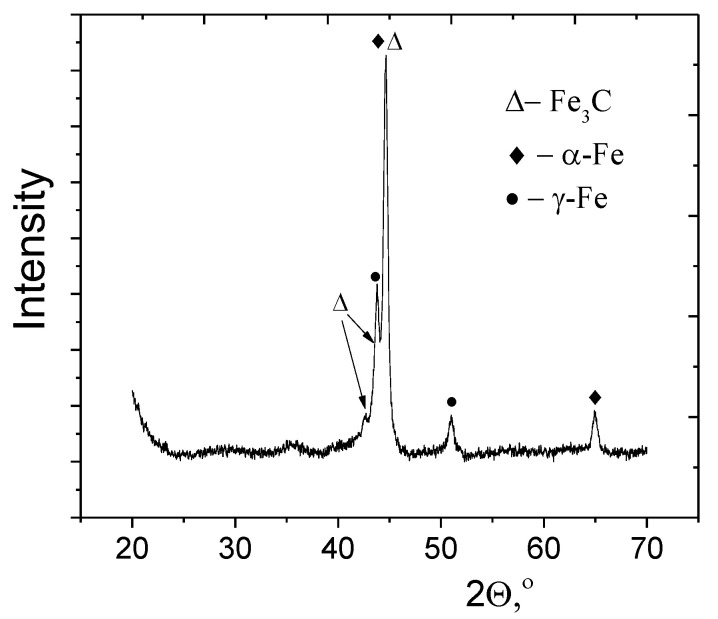
XRD pattern of produced Fe@C nanoparticles.

**Figure 3 nanomaterials-12-02869-f003:**
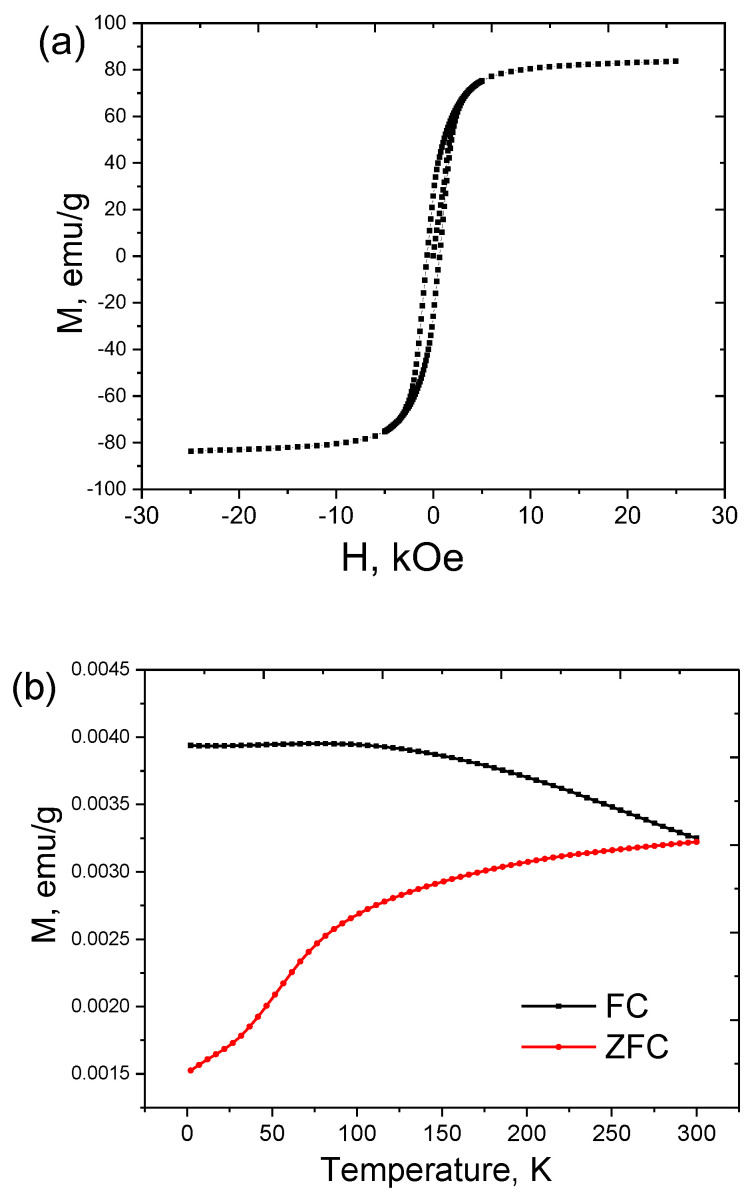
(**a**) Magnetization curve of Fe@C nanoparticles at room temperature; (**b**) zero-field-cooled (ZFC) and field-cooled (FC) magnetization curves measured under an applied magnetic field of 100 Oe.

**Table 1 nanomaterials-12-02869-t001:** Size and Zeta potential of BSA-Fe@C nanoparticles depending on Fe@C concentration.

Fe@C Concentration, mg/mL	Diameter, nm	Zeta Potential, mV
1.0	17 ± 3	−32 ± 5
2.5	22 ± 5	−32 ± 7
5.0	24 ± 7	−31 ± 7

**Table 2 nanomaterials-12-02869-t002:** Specific absorption rate (SAR) at different concentrations of Fe@C nanoparticles.

Concentration, mg/mL	SAR, W/g
1.0	437.64
2.5	129.36
5.0	50.4

**Table 3 nanomaterials-12-02869-t003:** SAR values comparison of our results with commonly used magnetic systems.

Nanoparticle	Concentration, mg/mL	*H* × *f,* A m^− 1^ s^− 1^	SAR, W/g	Reference
BSA-coated Fe@C	1	6.0 × 10^8^	437.64	This work
Aminosilane-coated NPs	5.0	1.4 × 10^10^	194.9	[17]
PVP–MNPs	1–10	5.0 × 10^9^	160	[60]

## Data Availability

Not applicable.

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
