# Peer review of "Albumin Stabilized Fe@C Core–Shell Nanoparticles as Candidates for Magnetic Hyperthermia Therapy"

_nanomaterials, 2022, doi:10.3390/nano12162869_

Round 1

Reviewer 1 Report

The authors reported the synthesis of BSA-coated Fe@C nanoparticles for hyperthermia applications. The work can be accepted after addressing the following comments:

1.  Novelty of this work is not very clear after reading the introduction. This should be highlighted in the introduction. 

2. Important reference related to the synthesis of crystalline Fe nanoparticles is missing,  Part. Part. Syst. Char. 2014, 31, 1054. 

3. From XRD data, there is a possibility for the oxidation of Fe iron nanoparticles.

4. "Increase of the concentration of the magnetic phase from 1 to 5 mg/ml leads to the increase of the nanoparticle diameter in the Fe@C-BSA system from 17 to 24 nm"- the increase in the size with concentration is not clear. Perhaps, it can also suggest the aggregation of nanoparticles. 

Author Response

Dear  Reviewer,

Reviewer 2 Report

Title: Albumin stabilized Fe@C core-shell nanoparticles as candidates for magnetic hyperthermia therapy

Authors: Ramírez-Morales et al.

Comments: This article studies Fe@C particles as hyperthermia seeds. The paper is well organized, but in my opinion it suffers from so many drawbacks that I cannot recommend its publication in its present form. Authors are requested to address the following questions.

Major Issues:

1. It is written that 1-5 mg of Fe@C powder was dispersed in 10 mL aqueous solutions. So, the final concentrations are 0.1, 0.25 and 0.5 mg/mL, not in the 1 to 5 mg/mL as claimed. Authors should correct/clarify this.

2. How did the authors estimate the specific heat capacity in J/gK units from Eq. 1 (and also note the error in line 123)?

3. They surmise that Ms ~84 emu/g is less than the saturation magnetization of Fe bulk due to the presence of small and superparamagnetic particles (which is not a reason for the diminished Ms), paramagnetic iron phases, and the diamagnetic carbon shell. In this sense, the reviewer points out that Figure 2 shows the presence of –mainly- Fe3C. Therefore, it is most likely that this is the reason for the measured value of Ms. The authors should comment on this [see, for instance, Hofer & Cohn, J. Am. Chem. Soc. 81, 1576 (1959) and Zhigach et al. Bull. Mater. Sci. 45, 38 (2022)]. Note that the bulk value for Fe3C is about 120 Am2/kg.

4. Figure 3 shows Hc ~620 Oe at RT, despite the ZFC-FC magnetization curves measured at 100 Oe collapse at 300 K, which means that there should be no irreversibility. Authors should comment on that.

5. The estimated SAR values (table 2) are intended to be directly proportional to the concentration of nanoparticles, which is odd. In principle, the SAR should not depend on the amount of material, unless the presence of magnetic dipole interactions modifies the magnetic response [see, Conde-Leboran et al. J. Phys. Chem. C 119, 15698 (2015)]. This is in addition to the problem mentioned in point 1 above. The reviewer believes that Eq. 1 has not been applied correctly. Authors should verify their conclusions.

6. Last but not least, Figure SI. 2 shows different experimental conditions during the acquisition of the heating curves (for example, the initial temperature ranges between ~19 or 21 Celsius). Authors should check/comment on this and also include an estimate of the error in their measurements.

Author Response

Dear Reviewer,

Round 2

Reviewer 2 Report

From my point of view, the article still suffers from serious flaws, as discussed below:

1). Equation 1 is not correct. Specific heat capacity has units of [J/gK] or [J/g°C] but not [J°C/g]. Also, it is written as (or similar)

SAR=(C/mMNP) ΔT/Δt, where, in adiabatic conditions, ΔT is the specimen temperature increment during the characterization time Δt, and C is assumed as Sci.mi, the specific heat capacity and the mass of each component in the sample (particles, dispersion medium and its container).

2) Despite the probable presence of iron phases, Figure 2 shows a peak due to Fe3C, which incidentally was not mentioned by the authors in their answer, and this being, from my point of view, the reason for the measured value of Ms. Indeed, a structure as Fe/Fe3C/C might very well happen. Cementite is a metallic ferromagnet and transforms to the paramagnetic state at a Curie temperature of 483 K. It also undergoes a phase transition of the internal magnetic field due to a spin-freezing process at about 75 K (see Guskos et al. J. Phys.: Conf. Series 10, 151 (2005), DOI: 10.1088 /1742-6596/10/1/037], that is, close to the 80 K point mentioned by the authors for magnetization bending in ZFC.

In addition, please see Wood, et al. J. Appl. Crystal. 37, 82 (2004), DOI: 10.1107/S0021889803024695, for the coefficient of volumetric thermal expansion as a function of temperature, which may shed light on the zfc-fc curves of Figure 3b. Authors should comment on that.

3) The x-scale in figure SI.2 should say "seconds" and not (ms). Also, are you sure that the linear fit parameters of temperature behavior over time in Table S1 are correct? That is, it seems that the expression y=5-5x+19.7 corresponds to the blue line (at 5 mg/mL instead of 1 mg/mL concentration) and vice versa (for y=1-5x+21.2 it corresponds to the line black at 1 mg/mL).

Author Response

Dear Reviewer,
